# Pharmacophore Mapping Combined with dbCICA Reveal New Structural Features for the Development of Novel Ligands Targeting α4β2 and α7 Nicotinic Acetylcholine Receptors

**DOI:** 10.3390/molecules27238236

**Published:** 2022-11-25

**Authors:** Victor S. Batista, Adriano Marques Gonçalves, Nailton M. Nascimento-Júnior

**Affiliations:** 1Laboratory of Medicinal Chemistry, Organic Synthesis and Molecular Modeling (LaQMedSOMM), Department of Biochemistry and Organic Chemistry, Institute of Chemistry, São Paulo State University (Unesp), Rua Professor Francisco Degni, 55, Jardim Quitandinha, Araraquara 14800-060, SP, Brazil; 2Department of Biological and Health Sciences, University of Araraquara (Uniara), Rua Carlos Gomes, 1217, Centro, Araraquara 14801-340, SP, Brazil

**Keywords:** pharmacophore mapping, dbCICA, drug design, α4β2 nAChRs, α7 nAChRs, molecular docking

## Abstract

The neuronal nicotinic acetylcholine receptors (nAChRs) belong to the ligand-gated ion channel (GLIC) group, presenting a crucial role in several biological processes and neuronal disorders. The α4β2 and α7 nAChRs are the most abundant in the central nervous system (CNS), being involved in challenging diseases such as epilepsy, Alzheimer’s disease, schizophrenia, and anxiety disorder, as well as alcohol and nicotine dependencies. In addition, in silico-based strategies may contribute to revealing new insights into drug design and virtual screening to find new drug candidates to treat CNS disorders. In this context, the pharmacophore maps were constructed and validated for the orthosteric sites of α4β2 and α7 nAChRs, through a docking-based Comparative Intermolecular Contacts Analysis (dbCICA). In this sense, bioactive ligands were retrieved from the literature for each receptor. A molecular docking protocol was developed for all ligands in both receptors by using GOLD software, considering GoldScore, ChemScore, ASP, and ChemPLP scoring functions. Output GOLD results were post-processed through dbCICA to identify critical contacts involved in protein-ligand interactions. Moreover, Crossminer software was used to construct a pharmacophoric map based on the most well-behaved ligands and negative contacts from the dbCICA model for each receptor. Both pharmacophore maps were validated by using a ROC curve. The results revealed important features for the ligands, such as the presence of hydrophobic regions, a planar ring, and hydrogen bond donor and acceptor atoms for α4β2. Parallelly, a non-planar ring region was identified for α7. These results can enable fragment-based drug design (FBDD) strategies, such as fragment growing, linking, and merging, allowing an increase in the activity of known fragments. Thus, our results can contribute to a further understanding of structural subunits presenting the potential for key ligand-receptor interactions, favoring the search in molecular databases and the design of novel ligands.

## 1. Introduction

The neuronal nicotinic acetylcholine receptors (nAChRs) belong to the ligand-gated ion channel (GLIC) group, which are involved in neurotransmitter release and excitation in the CNS [1]. They are expressed either in muscles or neuronal cells and are mostly expressed in the neuronal system in the case of the α4β2, α7, and α3β4 subtypes, which are the most abundant among all the nAChRs [2,3]. They are glycoproteins, presenting approximately 290 kDa and five transmembrane subunits, which are arranged around a cation permeable central pore. Their roles in several biological functions are related to the different types of structural combinations, involving α and β subunits, forming α homomeric structures or heteromeric ones, combining different α and β subunits, as the example of the α4β2 nAChRs, which can be found in two different stoichiometries: (α4)_2_(β2)_3_ and (α4)_3_(β2)_2_ [1,2,3].

Once nAChRs are highly diverse and involved in several biological processes, studies have shown their role in different neuronal disorders. The α7 nAChRs may be involved in epilepsy, Alzheimer’s disease, schizophrenia, and anxiety. On the other hand, α4β2 is related to epilepsy, Alzheimer’s disease, schizophrenia, Parkinson’s disease, and nicotine dependency [1,4,5]. In this sense, these receptors are promising targets to control and treat several neuronal disorders.

In fact, recent studies indicate that partial and full agonists of α7 nAChRs are effective in the treatment of neurological disorders such as schizophrenia and Parkinson’s disease and also in the control of drug-induced motor disorders, such as levodopa-induced dyskinesias which arises during the course of Parkinson’s disease treatment [6,7,8,9,10]. Concerning the α4β2 subtype, partial and full agonists have also been reported as promising ligands for conditions such as depression and nicotine addiction [11,12].

To comprehend ligands biding modes, it is necessary to analyze available nAChRs structures. In the case of the α4β2 receptor, the orthosteric binding site is located in the extracellular domain at the interface between the α4 and β2 subunits, the first face being called the principal face and the last face being the complementary face. The loops present on both subunits contribute to the formation of the binding site. Considering the interactions observed between nicotine (**1**) (Figure 1) and the α4β2 receptor (PDB ID 5KXI), key residues can be observed. For the α4 subunit, W156 contributes with cation-π, hydrogen bond, and hydrophobic interactions, Y204 with π-cation and hydrophobic, and residues T157, C200, and Y197 contribute by performing hydrophobic interactions. Residues L121 and W57 of the β2 subunit form hydrophobic interactions with nicotine [13]. In fact, the hydrophobic residues V111, F119, and L121 of the β2 subunit are related to the orientation of nicotine in the orthosteric site [14]. Considering the α7 nAChRs, each subunit contains a principal face and a complementary face. According to the crystallographic data of the humanized acetylcholine binding protein for α7 nAChRs (PDB ID 5AFH) [15], the key residues for the interactions involving lobeline (**2**) (Figure 1) on the principal face are W145 and Y184, forming interactions such as π-cation and hydrophobic, while Y191, C186, and C187 form hydrophobic interactions. For the complementary face, W53 and L116 form T-shaped and hydrophobic-π interactions, respectively.

In this context, computational studies capable of recognizing and assisting in the identification of relevant contacts, based on the correlation of molecular docking results and bioactivity data, are an excellent approach to search for ligands with high potential for interaction with the target protein. Therefore, the docking-based comparative intermolecular contacts analysis (dbCICA) methodology [16] is a viable strategy to guide the virtual screening of novel hits. The dbCICA is a methodology to identify the atoms in a given biding pocket that tend to interact with potent ligands, being also able to recognize those negative contacts, in other words, of atoms interacting with low-affinity ligands. In this way, it is possible to correlate docking poses with the bioactivity values of the ligands [17,18].

Regarding the search for novel bioactive compounds, computational chemistry has been used to optimize and accelerate drug development and design. Additionally, docking-based strategies, validation methods, pharmacophore mapping, and virtual screening have helped researchers to reduce invested time, labor, and the budget of the project. The pharmacophore model is a representation of the molecular structure and electronic features that are likely to interact with a determined target. After the pharmacophore mapping, it is possible to design molecules or select fragments and drugs by virtual screening, followed by in vitro assessment, accelerating the process of drug discovery [19].

In this sense, the aim of this research was to construct and validate pharmacophore maps for the orthosteric sites of (α4)_2_(β2)_3_ and α7 nAChRs, by using dbCICA methodology, to bring up new insights in drug search and design for these targets.

## 2. Results and Discussion

The 10 dbCICA models with the highest r^2^ 5-fold for (α4)_2_(β2)_3_ are described in Table 1. The critical contacts according to the two best models are listed in Table 2.

According to Table 1, the best docking condition occurred considering ChemScore function and protonated ligands at pH 7.4, a water molecule in the active site and maximum contact distance equal to 3.5 Å, resulting in an r^2^ 5-fold of 0.593 and F-statistic of 157.21. The r^2^ 5-fold value indicates that approximately 59% of the observed variance for Y was explained through the regression model. It is understood that this value is acceptable since the regression model correlates bioactivity, a complex property with the sum of inverted positive and negative contacts, a single descriptor, and high simplicity. Additionally, the aim of the dbCICA analysis is to identify which contacts are determinants for bioactivity and to use this information for virtual screening so that the prediction of activity values is only a tool for building and ranking the models and not one of the properties to be evaluated. As for the F statistic for the nAChRs_α4β2_1 model, the critical F value for the model’s degrees of freedom (1 and 96) is equal to 3.94, with *p* = 6.2 × 10^−22^, indicating good fitting.

Ligands ionization, compared to the presence or absence of the water molecule, seems to be more critical for analysis since, for five of the ten models, the water is not present, and only three of the ten models have non-ionized ligands. For the contacts indicated by the two best models (Table 2), there is an agreement regarding the contact between the carbon atom of residue A:W156, reinforcing its importance. Such contact represents a hydrogen bond between the hydrogen atom bound to the basic nitrogen atom of the ligand and the carbonyl group located in the A:W156 residue, in agreement with several research groups that reported the importance of this interaction for molecular recognition of ligands targeting the α4β2 nAChRs [20,21,22,23]. The contact of the CG atom located in residue A:Y204 (Figure 2a) in the model nAChRs_α4β2_1 represents hydrophobic interactions as a result of the A:W156 hydrogen bond, as it tends to orient the α-N carbon atoms towards this residue. Furthermore, when positively charged regions of the ligand participate through a hydrogen bond, cation-π interactions occur involving the aromatic residues of this region. Published studies indicate that the cation-π interaction between ligands, such as acetylcholine, nicotine (**1**), and carbamylcholine, involving the side chain of residue W156 is essential for the activity of these molecules [23,24]. The proximity of active ligands to the atom HG23 of residue A:T157 (Figure 2a) occurs due to the presence of the water molecule in the active site, as it tends to maximize the interactions between the heteroaromatic rings and HG23. Some negative contacts pointed out by the dbCICA model are also supported in the literature. For example, it is known from site-directed mutagenesis studies that the Y100F mutation reduces the affinity of acetylcholine and carbamylcholine since phenylalanine residue does not have a hydroxyl group, which is important for the interaction of these ligands [25,26]. Similarly, mutations in Y197 reduce the affinity of acetylcholine for the receptor [26,27]. Therefore, such contacts are justified since they represent steric shocks with residues that are decisive for the biological activity of the ligands involved. There is good agreement between the results obtained from the dbCICA nAChRs_α4β2_1 model and the literature data, reinforcing its validity.

The overlay shown in Figure 2 reveals common regions for the most well-behaved ligands that were used in the construction of the pharmacophore map presented in Figure 3. A characteristic sphere was created for positively charged atoms where there was an overlap of basic functional groups, also representing the contact with A:Y204:CG because this interaction is implicated in the proximity of the α-N atoms of the ligands with A:Y204 residue. In this same region, a hydrogen bond donor vector oriented in the direction of A:W156:C was modeled. The contact with A:T157:HG23 is represented by a sphere of hydrogen bond acceptor because it is believed that the proximity to this atom is due to the water molecule in the binding site. Additionally, close to this point, an aromatic ring sphere was built where this type of subunit overlapped. The contact with B:F119:CB is represented by a hydrophobic sphere. This sphere is not positioned in an atom once this contact occurs due to the interaction with the phenyl ring of the residue.

Meshed spheres are shown as follows: exclusion spheres in gray; hydrogen bond acceptor regions in red; planar rings in light green; hydrophobic regions in yellow; hydrogen bond donor regions in blue. Blue opaque spheres represent the directions of hydrogen bond donor atoms.

Nitrogen atoms are shown in blue, oxygen atoms in red, fluor atoms in light blue, hydrogen atoms are omitted for clarity and carbon atoms in gray (amino acids) or black (ligand). Molecular interactions are shown as dotted lines: hydrogen bonds in green, π-anion in dark blue, π-sigma in cyan, π-π stacked in pink, and π-alkyl in black. 

According to Scott et al. (2012), three strategies can be used to increase the efficiency of known ligands on a given target: fragment growing, fragment linking, and fragment merging. The fragment growth starts from a fragment that must be incremented with substructures that have the potential to interact with additional target regions. Fragment linking aims to link two fragments with known interactions in different regions of the binding site in order to take advantage of the maximum potential of interaction based on the characteristics of the original ligands. On the other hand, the fragment merge strategy aims to join fragments based on the overlapping substructures, preserving the interaction core and exploring the involved side chains focusing on a greater interaction potential [28].

Based on these strategies, it is possible to design novel structures derived from known active molecules (Figure 4). Most of the well-behaved (α4)_2_(β2)_3_ ligands occupy a large volume of the binding pocket, in this sense, an alternative to increased activity through fragment merging strategy (Figure 4, Figure 5 and Figure 6). In addition, pyridine bounded to cyclopropane substructure, present in ligands **6**, **7**, **9**, **10**, **14**, and **15** (Figure 4), can be used as a merging point to obtain novel structures presenting higher activity potential. One possibility is merging **6** and **16**, also considering the new isomers, selecting the azacyclopentane from **6** and the carbon chain with the methoxy group from **16**.

Regarding the α7 subtype, all 93 structures were docked in the receptor according to the described methodology in all the mentioned docking conditions. Using the poses related to the best-scored enantiomers (asymmetric atoms can be nitrogen or carbon) for each ligand, several dbCICA models were generated as previously described. The 10 models presenting the highest r^2^ 5-fold are described in Table 3. The critical contacts pointed out by the two best models are listed in Table 4.

**Figure 4 molecules-27-08236-f004:**
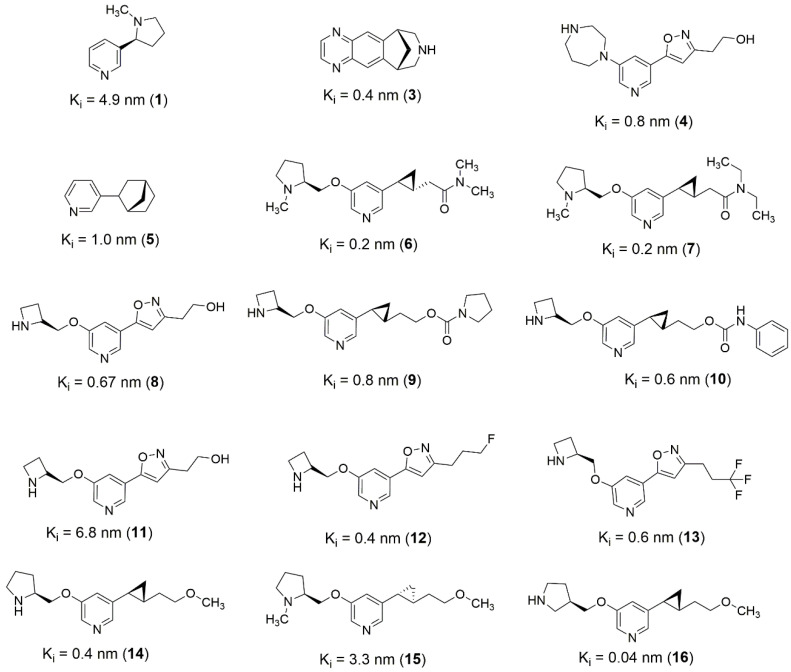
2D structures of best well-behaved ligands used for (α4)_2_(β2)_3_ receptor pharmacophore mapping.

**Figure 5 molecules-27-08236-f005:**
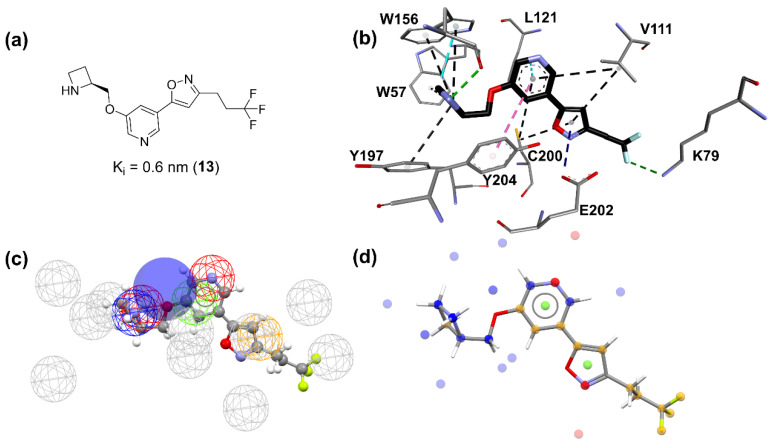
Characteristics and docking pose of ligand **13**. (**a**) 2D structure, (**b**) best docking pose and its interactions with (α4)_2_(β2)_3_ orthosteric site, (**c**) **13** structure in pharmacophore map, (**d**) **13** structural and molecular features. (**b**) Nitrogen atoms are shown in blue, oxygen atoms in red, fluorine atoms in light blue, hydrogen atoms are omitted for clarity and carbon atoms in gray (amino acids) or black (ligand). Molecular interactions are shown as dotted lines: hydrogen bonds in green, π-anion in dark blue, π-sigma in cyan, π-π stacked in pink, and π-alkyl in black. (**c**) Meshed spheres are shown as follows: exclusion spheres in gray; hydrogen bond acceptor regions in red; planar rings in light green; hydrophobic regions in yellow; hydrogen bond donor regions in blue. Blue opaque spheres represent directions of hydrogen bond donor atoms. (**d**) Small spheres represent molecular features and are shown as follows: proton acceptor regions in red; planar rings in light green; hydrophobic regions in yellow; hydrogen bond donor regions in blue.

**Figure 6 molecules-27-08236-f006:**
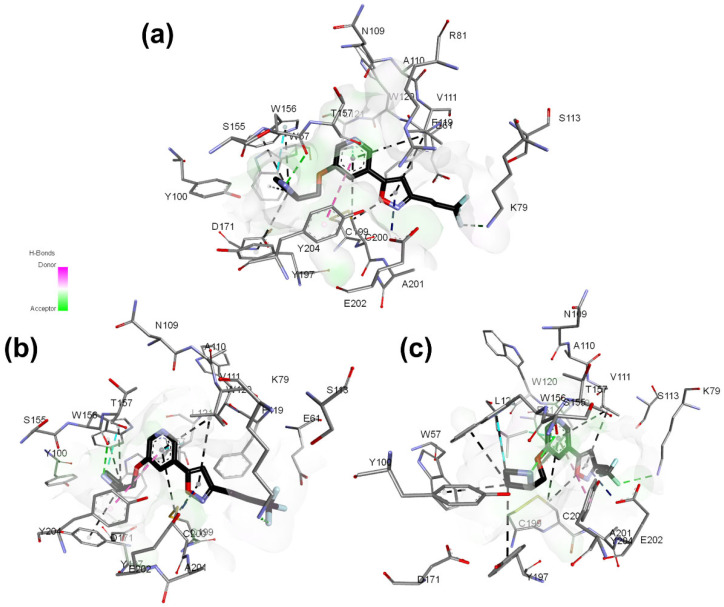
Best docking pose of **13** and hydrogen bond potential surface of (α4)_2_(β2)_3_ receptor binding pocket. (**a**–**c**) correspond to different site views.

**Table 3 molecules-27-08236-t003:** Docking conditions and statistics for the 10 best dbCICA models for α7 receptor.

nAChRs Models	Score Function	Water	Ionization	Contacts Distance (Å)	Contacts *	r^2^ 5-Fold	F-Statistics
α7_1	ChemPLP	No	Yes	2.5	3_10	0.582	134.37
α7_2	ChemPLP	No	Yes	3.5	3_10	0.534	118.08
α7_3	ChemPLP	No	Yes	2.5	10_10	0.530	114.28
α7_4	ChemPLP	Yes	Yes	2.5	6_5	0.529	107.08
α7_5	ChemPLP	No	Yes	3.5	3_5	0.529	112.61
α7_6	ChemPLP	No	Yes	2.5	8_10	0.523	104.75
α7_7	ChemPLP	No	Yes	2.5	7_5	0.521	108.08
α7_8	ChemPLP	Yes	Yes	2.5	8_10	0.516	102.42
α7_9	ChemPLP	Yes	Yes	2.5	5_10	0.514	100.67
α7_10	ChemPLP	No	Yes	2.5	5_10	0.511	101.92

* The first number is related to positive contacts and the second to negative contacts.

**Table 4 molecules-27-08236-t004:** Critical contacts for the two best dbCICA models for α7 receptor.

Model	Positive Contacts (Weight)	Negative Contacts
nAChRs_α7_1	B:Y184:CD1 (2), B:Y191:CG (2), B:Y191:HD2 (3)	C:A105:HA, C:L106:HG, B:W145:HA, C:W53:CZ2, B:Y184:CE1, B:Y184:CZ, B:Y184:HB2, B:Y191:CD1, B:Y191:HH, B:Y91:HB1
nAChRs_α7_2	B:K141:NZ (3), B:S144:O (2), B:Y184:HB2 (2)	C:A105:CA, B:C186:HB1, B:C187:SG, C:L106:CA, B:W145:CA, C:W53:CZ3, C:Y115:N, B:Y91:C, B:Y91:CA, B:Y91:CE1

Note: The atoms were automatically named by the Discovery Studio Visualizer software, and the residue numbering respects the originally published numbering from PDB structure.

According to the data presented in Table 3, the best docking condition to explain the variation in the bioactivity of the training group model considers ChemPLP function with protonated ligands at pH 7.4, without the water molecule in the active site and maximum contact distance of 2.5 Å, resulting in an r^2^ 5-fold equal to 0.582 and F-statistic value of 134.37. r^2^ 5-fold statistic indicates that approximately 58% of the variance observed for Y was explained through the regression model. The model is acceptable and has a similar r^2^ 5-fold to the analysis of the nAChRs_α4β2_1 model. As well as for the F-statistic of the nAChRs_α7_1 model, the critical F-value for the model’s degrees of freedom (1 and 93) is equal to 5.11, with *p* = 1.29 × 10^−13^, indicating a good fit.

In the case of the models built for the α7 receptor, all 10 best-calculated models used the ChemPLP function, indicating its efficiency for this case. The ionization state of the ligands was very important since it is observed for all 10 models described in Table 3. The presence of water was not critical for molecular recognition since seven out of ten models did not use it during the calculation. The analysis of the critical contacts pointed out that model nAChRs_α7_1 follows the same premises of the analysis for the model nAChRs_α4β2_1: The positive contacts shown represent interactions involving the residues of the aromatic pocket in the active site, where positively charged nitrogen atoms of the ligands can participate in cation-π interactions involving such residues (Figure 7). Furthermore, the same residues can participate in T-stacking and π-stacking aromatic interactions when the ligand has an aromatic moiety, which is present in several ligands presenting activity in nAChRs.

The nAChRs_α7_1 model does not present critical contacts involving the carbonyl group of the residue B:W145, while the contact involving A:W156:C of the model nAChRs_α4β2_1 is observed. This tryptophane residue is highly conserved among the nAChRs [29]. This finding corroborates data from other studies, which indicate that the mechanism by which the selectivity of α4β2 nAChRs, in contrast with the α7 subtype, occurs due to the formation of a hydrogen bond with tryptophan residue in α4β2 receptor and the absence of it for the α7 receptor. Due to the presence of a lysine residue in a region bordering the active site, hydrogen bonds are formed, positioning the tryptophan (W156) of the α4β2 subtype in such a way that the hydrogen bond interaction is favored. For the α7 subtype, the lysine is replaced by glycine; molecular dynamics and site-directed mutagenesis studies indicate that this substitution causes the positioning of the tryptophan (W145) to be unfavorable for the formation of the hydrogen bond [2,30,31]. In this sense, the model was able to indicate critical interactions well documented in the scientific literature, corroborating the selectivity patterns already established for these two receptors.

The pharmacophore map for the α7 receptor (Figure 8) was constructed with information from the well-behaved ligands (Figure 9) and negative contacts from the best dbCICA model (Figure 7). In this sense, a non-planar ring sphere was positioned at the site of overlapping for the bicyclic moieties of the ligands since this substructure can form π-alkyl interactions with residues B:Y91, B:W145, B:Y184, and B:191. Additionally, two hydrogen bond donor spheres were positioned close to this region because they favor non-classical and electron-deficient atoms’ hydrogen bonds. In addition to the non-planar ring region, a hydrophobic sphere was added due to the hydrophobic characteristic of the region related to the non-planar and planar rings of the ligand; next, a planar ring sphere was also added. A hydrogen bond acceptor sphere was positioned between the oxygen and nitrogen atoms present in most of the well-behaved ligands planar rings. Then, a hydrophobic sphere was added due to the hydrophobic properties present in most well-behaved ligands. Furthermore, exclusion spheres were modeled on the atoms identified as negative contacts based on the best dbCICA model (Table 4).

Based on Figure 9, Figure 10 and Figure 11, it is possible to explore fragment growth strategies for the well-behaved ligands, analyzing their characteristics along the binding site. The docking pose indicates that the best region for fragment growth approach is opposite to the bicycle, present in most ligands (Figure 9), since it is properly positioned, with a good occupation of the binding pocket volume. Therefore, based on the characteristics of the amino acid residues around the opposite side of the bicycle, it is possible to grow the structure with the insertion of groups that form hydrogen bonds, such as amines or amides, favoring possible interactions with the Y91, S118, and G163 residues. In addition, in the same region, the insertion of nonpolar groups in the ligand, such as phenyl (aromatic) and t-butyl (alkyl) groups, could allow, for instance, T-stacking and alkyl-alkyl interaction with W53 and L36 residues.

The ROC curve for the validation of nAChRs_α7_1 and nAChRs_α4β2_1 dbCICA models is available as Appendix A. Both models were suitably validated with AUC of 0.93 and 0.86, AAC of 0.97 and 0.90, and YA of 0.26 and 0.25 for nAChRs_α7_1 and nAChRs_α4β2_1, respectively. Therefore, the pharmacophoric maps are reliable for virtual screening and fragment-based drug design strategies.

## 3. Materials and Methods

The crystal structures of (α4)_2_(β2)_3_ (PDB ID 5KXI, 3.9 Å resolution) and α7 (PDB ID 5AFH, 2.4 Å resolution) nAChRs subtypes were obtained from the Protein Data Bank, containing nicotine (**1**) and lobeline (**2**) as co-crystalized ligands, respectively (Figure 1). Indeed, the 5AFH PDB structure is a chimeric protein between the AChBP from *Lymnaea stagnalis* species and the extracellular domain of the human α7 nAChRs, presenting 71% similarity to the native human protein. The overall structure and orthosteric site quality were analyzed through the PROCHECK tool within the SAVES server [32]. The choice of the 3D structures for the nAChRs was based on those available during the design and development of the study, being 5KXI for α4β2 nAChRs and the humanized acetylcholine binding protein 5AFH for α7 nAChRs. Regarding α4β2 nAChRs (PDB ID 5KXI), more recent structures do not differ significantly in terms of resolution, such as 6CNK (3.90 Å), 6CNJ (3.70 Å) and 6UR8 (3.71 Å). For the α7 nAChRs, despite the cryoEM 7KOX structure having a better resolution (2.70 Å), the humanized α7-AChBP (PDB ID 5AFH) structure has good global structural overlap (RMSD 1.757), with RMSD of 1.198 for the following key residues of the orthosteric site: B:W145, B:Y184, B:C186, B:C187, B:Y191, C:W153, and C:L116 (the letters correspond to the chain in the PDB file and the residue numbering is in accordance with the 5AFH structure).

The virtual screening validation was performed through the docking-based Comparative Intermolecular Contacts Analysis (dbCICA) [16]. A set of molecules with known bioactivities for each subtype was used to validate the computational methods, where the molecules were selected based on pharmacological assays presenting measured affinity (Ki) through the same protocol. In this sense, 98 molecules showing Ki between 0.04 and 2176 nM were selected to compose the training group of (α4)_2_(β2)_3_ subtype model (Appendix A). For the construction of the α7 subtype model, it was necessary to use pairs of non-interconvertible enantiomers for some molecules in order to increase the training set, and the enantiomer selected for model construction was the one presenting the highest docking score. In this sense, 93 molecules, presenting Ki between 5.4 nM and 2180 μM, were obtained to compose the training group of α7 subtype model (Appendix A). Ki values were converted to pKi (−log(Ki)) to linearly correlate ligand bioactivity with a variation of free energy [16].

The three-dimensional structures of the ligands were built in silico, with explicit atoms, using Discovery Studio Visualizer software (v.17.0.2.1076) [33]. The protonation states of the molecules were determined in pH = 7.4 [34]. Ligands defined as protonated at the chosen pH were constructed in the two possible enantiomeric forms in comparison with the basic nitrogen, when applicable. The structures were energetically minimized through the PM7 semi-empirical method by using the graphical interface of Mercury CSD (v.3.9) [35] while the calculation was performed with MOPAC2016 software [36].

Molecular docking was performed with the GOLD suite software (V 5.4) [35]. Both nAChRs structures were prepared for docking by removing co-crystalized ligands and adding hydrogen atoms. The center of the search site radius for the (α4)_2_(β2)_3_ model was defined at the coordinates x = 67.3680, y = −27.2034, and z = −39.0781; this being the position of the pyrrolidine nitrogen atom of co-crystalized nicotine. For the α7 model, the center of the search site radius was defined at coordinates x = −20.8987, y = −10.0015, and z = 8.5084, which is the position of the basic nitrogen atom of the co-crystalized lobeline. The search radius for both models was set as 10 Å, and rigid docking was performed, considering 50 docking poses for each ligand. For the (α4)_2_(β2)_3_ model, a molecule of water was modeled in the active site region at coordinates x = 69.3500, y = −22.8900, and z = −42.4450, in accordance with published works [20,37]. Regarding the α7 subtype model, a water molecule was modeled at coordinates x = −17.5010, y = −12.1430, and z = 5.4700, corresponding to the position of a co-crystallized water molecule. The water molecule had free rotation and translational motion of up to 2 Å in the x, y, and z axes in both models. For ligands flexibility, free rotation for single bonds, inversion of planar amines and amides by 180°, as well as carboxylic acids, and variations in the conformations of non-aromatic rings were allowed. For the construction of dbCICA models, the higher scoring enantiomeric forms were selected in cases where there were two possibilities for ionization in basic nitrogen atoms and, regarding the α7 model, among carbon enantiomers pairs, the highest scored structure was selected analogously, when applicable.

Concerning dbCICA validation, rigid docking was performed for 98 and 93 known bioactive molecules, for (α4)_2_(β2)_3_ and α7, respectively, in both protonated and deprotonated forms, in the presence and absence of a water molecule in the active site as well as using the four scoring functions of the GOLD program: ASP, ChemScore, GoldScore, and ChemPLP, resulting in a total of 16 docking runs for each nAChRs in silico model. The genetic algorithm of dbCICA was implemented with the MatLab software (v.R2007b) [38], and the r^2^ 5-fold correlation coefficient was chosen as the fitting method for both models [16]. At the end of the iterations of the genetic algorithm, the model with the highest r^2^ 5-fold was defined as the best dbCICA model.

Microsoft^®^ EXCEL regression tool (365 ProPlus) was used to correlate the sum of contacts indicated by the best dbCICA model and the experimental bioactivity of the ligands to predict the values of pKi for each ligand. Then, the most active and well-behaved ligands within the model were selected as they have the lowest absolute residual difference between the real and predicted value. The docking poses of the referred ligands were superimposed on the receptor’s active site within the CSD CrossMiner software environment (v. 1.4). In this sense, the pharmacophore map for both models was constructed based on ligands structures and negative contact atoms, pointed by dbCICA analysis. The tolerance radius allowed for the exclusion and feature spheres was 1.6 Å, with the exception of the feature sphere, which represents the spatial projection associated with hydrogen bond donors whose radius was 2.2 Å.

The pharmacophoric maps were validated through Receiver Operating Characteristic Curve (ROC). In this way, 34 and 25 of the known active ligands were chosen to select decoys from ZINC and ChEMBL databases for (α4)_2_(β2)_3_ and α7 models, respectively. Regarding the choosing criteria of ligand for validation, all known active ligands with Euclidian distance, different from zero, were selected (see Appendix A), then the 30% most potent, with different Ki, were chosen. A decoy search was conducted with the software Decoyfinder 2.0 [39], and 25 decoys were selected for each active ligand, with a total of 850 presumably inactive substances for (α4)_2_(β2)_3_ and 625 for α7. The search parameters were set to minimize the artificial enrichment of the models [40]. In this sense, Tanimoto’s similarity between actives and inactives was at least 0.85, and the similarity between inactives and the other 24 inactives in the set for the same group was also at least 0.85, assuming that the inactives had ±1 donor and ±1 hydrogen bond acceptor in relation to their respective actives. The molecular weight of the inactives was allowed to vary by ±48 Da in relation to their actives, ±2 rotatable bonds were also accepted, the water-octanol partition coefficient (logP) could vary by ±1 and the maximum standard deviation of inactives in comparison to their actives was 1.5. MarvinSketch was used to predict the major ionization microspecies of the molecules at 7.4 pH. Then, Mercury software was used to generate up to 500 conformers for each molecule, including actives and inactives. A database for each set of molecules, considering independent databases for (α4)_2_(β2)_3_ and α7 subtype models, was constructed with CSD CrossMiner. A hit search was performed with the same software. The results were analyzed, and the ROC curve was determined through OriginPro (v.8) [41].

## 4. Conclusions

It was possible to build up and validate pharmacophoric maps for the orthosteric sites of (α4)_2_(β2)_3_ and α7 nAChRs through a dbCICA approach. In this sense, the models were able to reproduce critical nAChRs-ligand interactions well documented in the scientific literature, corroborating the selectivity patterns already established for these two receptors. Thus, our results contribute to the understanding of structural features of the orthosteric sites of (α4)_2_(β2)_3_ and α7 nAChRs. Our data reveal important structural characteristics to develop and search for novel ligands to find substances with higher activities. Additionally, our computational approach leads to promising future studies involving virtual screening in molecular databases for the design of bioactive compounds targeting (α4)_2_(β2)_3_ and α7 nAChRs. Additionally, this approach may be used with different targets, speeding up the search for novel ligands and decreasing research and development costs.

## Figures and Tables

**Figure 1 molecules-27-08236-f001:**
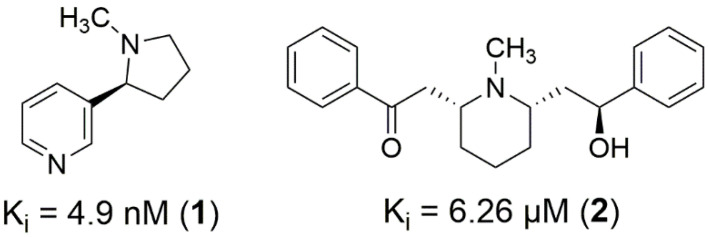
2D structures of nicotine (**1**) and lobeline (**2**), followed by K_i_ values for (α4)_2_(β2)_3_ and α7 nAChRs subtypes, respectively.

**Figure 2 molecules-27-08236-f002:**
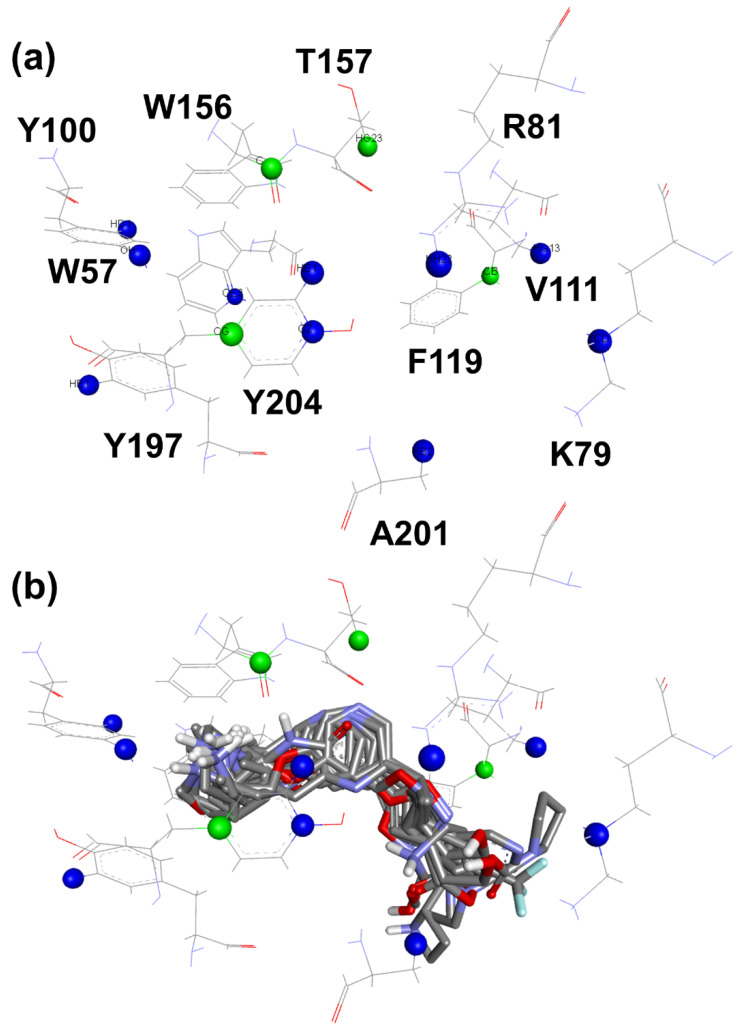
(**a**) Critical contacts from (α4)_2_(β2)_3_ receptor dbCICA analysis and (**b**) best well-behaved ligands from dbCICA analysis overlay. (**a**) Positive contacts from (α4)_2_(β2)_3_ receptor dbCICA model are shown in green and negative contacts are shown in blue. (**b**) Superposition of the best well-behaved ligands from dbCICA analysis (see ligands **1**, **3**, **4**, **5**, **6**, **7**, **8**, **9**, **10**, **11**, **12**, **13**, **14**, **15,** and **16**, in Figure 4). Carbon atoms are shown in gray, nitrogen atoms in blue, oxygen atoms in red, fluorine atoms in light blue and hydrogen atoms in white. The Discovery Studio Visualizer software automatically named the atoms, and the residue numbering respects the originally published numbering from the PDB structure. Only polar hydrogens from ligands are shown.

**Figure 3 molecules-27-08236-f003:**
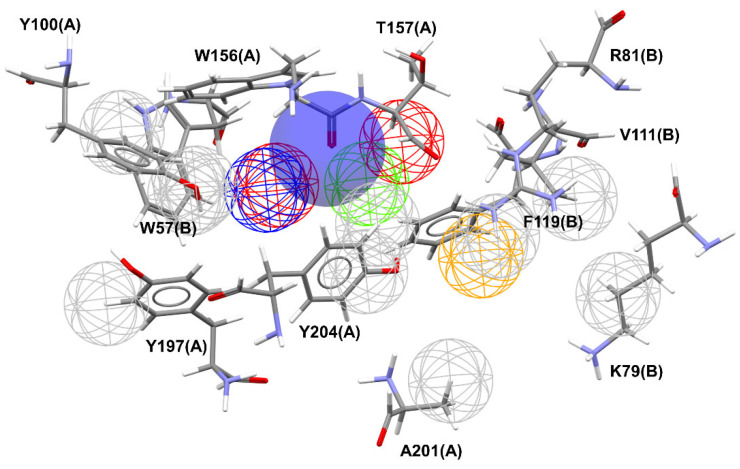
Pharmacophore map of the best well-behaved ligands from (α4)_2_(β2)_3_ receptor dbCICA model combined with exclusion spheres from (α4)_2_(β2)_3_ receptor. The letters correspond to the chain in the PDB file and the residue numbering is in accordance with the 5KXI structure).

**Figure 7 molecules-27-08236-f007:**
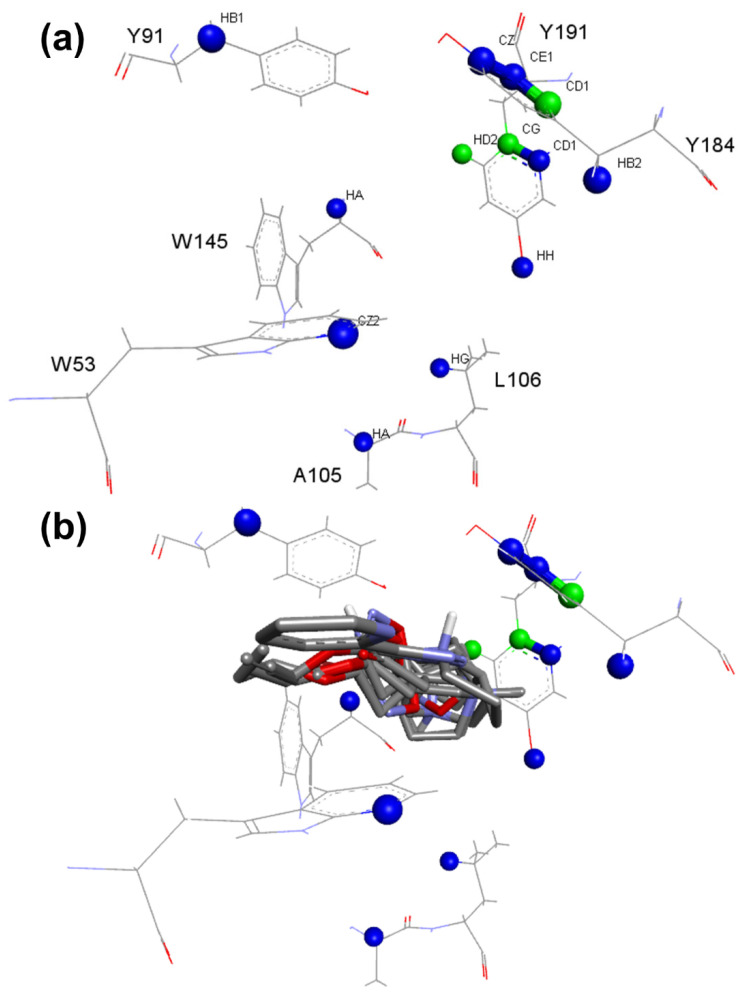
Critical contacts from α7 receptor dbCICA analysis (**a**) and most well-behaved ligands from dbCICA analysis overlay (**b**). (**a**) Positive contacts from α7 receptor dbCICA model are shown in green and negative contacts are shown in blue. (**b**) Superposition of the best well-behaved ligands from dbCICA analysis (see ligands **2**, **17**, **18**, **19**, **20**, **21**, **22**, **23**, and **24** in Figure 9). Carbon atoms are shown in gray, nitrogen atoms in blue, oxygen atoms in red and hydrogen atoms in white. The atoms were automatically named by the Discovery Studio Visualizer software, and the residue numbering respects the originally published numbering from PDB structure. Only polar hydrogens from ligands are shown.

**Figure 8 molecules-27-08236-f008:**
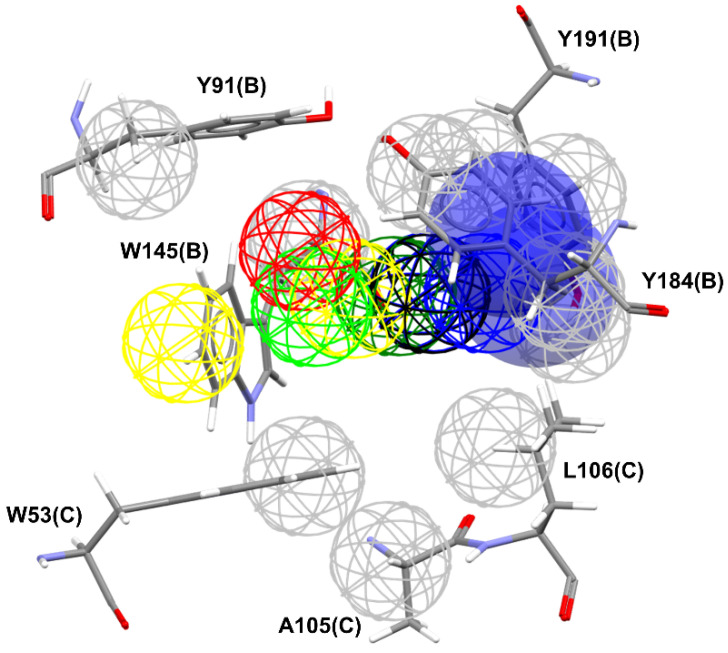
Pharmacophore map of the most well-behaved ligands from α7 receptor dbCICA model combined with exclusion spheres from α7 receptor. Meshed spheres are shown as follows: exclusion spheres in gray; hydrogen bond acceptor regions in red; non-planar rings in dark green; planar rings in light green; hydrophobic regions in yellow; positively charged regions in black; hydrogen bond donor regions in blue. Blue opaque spheres represent directions of hydrogen bond donor atoms. Hydrogen bond donor regions include non-classical and electron-deficient atoms of hydrogen bonds.

**Figure 9 molecules-27-08236-f009:**
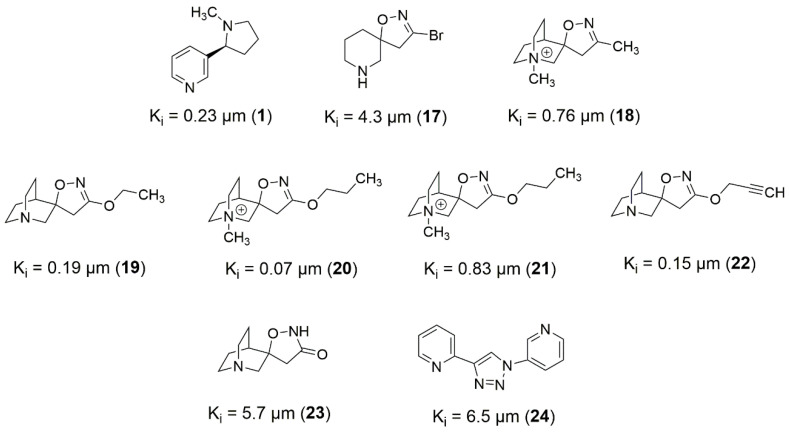
2D structures for the most well-behaved ligands used for α7 receptor pharmacophore mapping.

**Figure 10 molecules-27-08236-f010:**
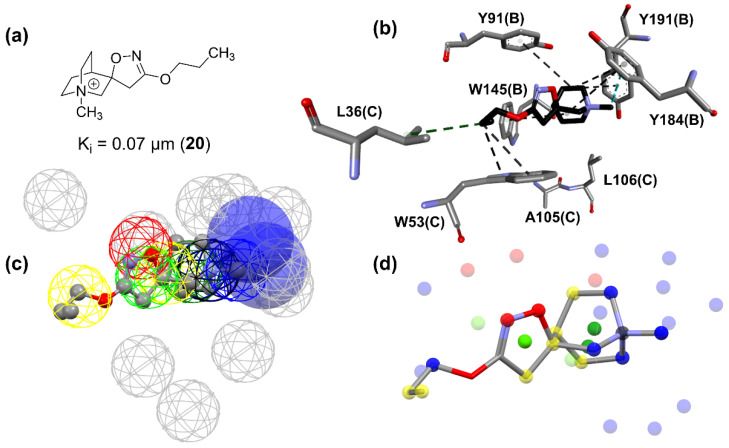
Characteristics and docking pose of ligand **20**. (**a**) 2D structure, (**b**) best docking pose and its interactions with α7 orthosteric site, (**c**) **20** structure in pharmacophore map, (**d**) structural and molecular features. (**b**) Nitrogen atoms are shown in blue, oxygen atoms in red; hydrogen atoms were omitted for clarity and carbon atoms in gray (amino acids) or black (ligand). Molecular interactions are shown as dotted lines: alkyl-alkyl in green, π-sigma in cyan, and π-alkyl in black. (**c**) Meshed spheres are shown as follows: exclusion spheres in gray; hydrogen bond acceptor regions in red; non-planar rings in dark green; positively charged regions in black; planar rings in light green; hydrophobic regions in yellow; hydrogen bond donor regions in blue. Blue opaque spheres represent directions of hydrogen bond donor atoms. (**d**) Small spheres represent molecular features and are shown as follows: proton acceptor regions in red; planar rings in light green; non-planar rings in dark green; hydrophobic regions in yellow; positively charged regions in black; hydrogen bond donor regions in blue.

**Figure 11 molecules-27-08236-f011:**
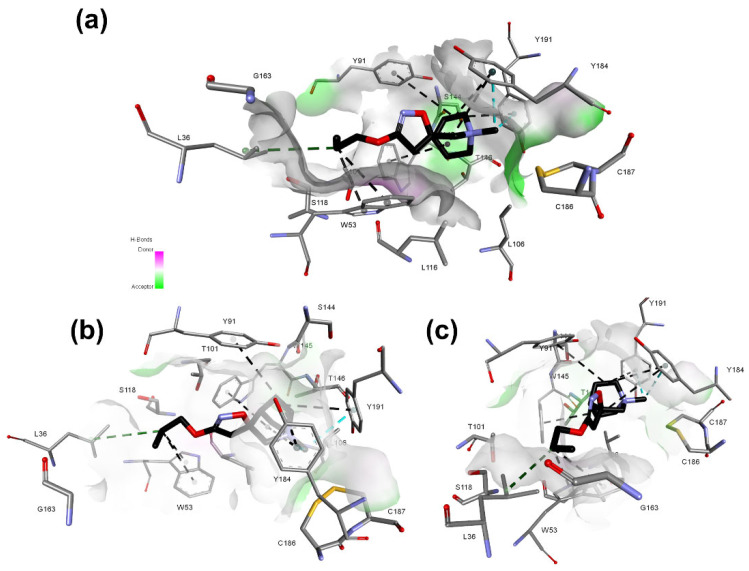
Best docking pose of **20** and hydrogen bond potential surface of α7 receptor binding pocket. Nitrogen atoms are shown in blue, sulfur in yellow, oxygen atoms in red, hydrogen atoms were omitted for clarity and carbon atoms in gray (amino acids) or black (ligand). Molecular interactions are shown as dotted lines: alkyl-alkyl in green, π-sigma in cyan, and π-alkyl in black. (**a**–**c**) correspond to different site views.

**Table 1 molecules-27-08236-t001:** Docking conditions and statistics for the 10 best dbCICA models for (α4)_2_(β2)_3_ receptor.

Model	Score Function	Water	Ionization	Contacts Distance (Å)	Contacts *	r^2^ 5-Fold	F-Statistics
nAChRs_α4β2_1	ChemScore	Yes	Yes	3.5	4_10	0.593	157.21
nAChRs_α4β2_2	ASP	Yes	No	3.5	3_10	0.592	146.23
nAChRs_α4β2_3	ChemPLP	No	No	3.5	4_10	0.582	139.92
nAChRs_α4β2_4	GOLDScore	No	Yes	3.5	4_10	0.573	134.58
nAChRs_α4β2_5	ChemScore	Yes	Yes	3.5	3_10	0.568	141.80
nAChRs_α4β2_6	ChemPLP	No	Yes	3.5	4_5	0.565	127.31
nAChRs_α4β2_7	ChemPLP	No	Yes	3.5	3_10	0.563	136.79
nAChRs_α4β2_8	GOLDScore	No	Yes	2.5	4_10	0.555	127.76
nAChRs_α4β2_9	ChemPLP	Yes	Yes	3.5	3_10	0.555	125.06
nAChRs_α4β2_10	ASP	Yes	No	3.5	4_10	0.549	125.65

* The first number is related to positive contacts and the second to negative contacts.

**Table 2 molecules-27-08236-t002:** Critical contacts for the two best dbCICA models for (α4)_2_(β2)_3_ receptor.

Model	Positive Contacts (Weight)	Negative Contacts
nAChRs_α4β2_1	B:F119:CB (2), A:T157:HG23 (2), A:W156:C (3), A:Y204:CG (2)	A:A201:HB2, B:R81:HH22, B:K79:CD, B:W57:CE3, A:Y100:HD1, A:Y100:OH, A:Y197:HE1, A:Y204:CZ, A:Y204:HE1, B:V111:HG13
nAChRs_α4β2_2	HOH:1 (2), B:L121:HG (2), A:W156:C (2)	A:C200:HB32, B:L121:CD1, B:F119:HD2, B:F119:HE2, B:F119:HZ, A:W156:CB, A:W156:CG, A:W156:HH2, B:W57:CE2, A:Y204:HE2

Note: The atoms were automatically named by the Discovery Studio Visualizer software, and the residue numbering respects the originally published numbering from PDB structure.

## Data Availability

Not applicable.

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
