# Peer review of "Pharmacophore Mapping Combined with dbCICA Reveal New Structural Features for the Development of Novel Ligands Targeting α4β2 and α7 Nicotinic Acetylcholine Receptors"

_molecules, 2022, doi:10.3390/molecules27238236_

Round 1

Reviewer 1 Report

This is an interesting study conducted by new scientists in the field of nicotinic acetylcholine receptors, as deduced by their record in Scopus. The presented study is strictly computational combining a number of tools to identify contacts that influence protein-ligand interactions and to construct a pharmacophoric map.  

Although the whole study is descent in all aspects, the results and the novelty are not presented well. The manuscript is missing a description of the orthosteric binding site, which is the main object of the study. There are several crystallographic and cryo-EM structures revealing the similarities and distinct features of the binding sites among the various nAChR subtypes and the authors essentially do not comment nothing of those. The readers should have been informed early on about the importance of the critical residues discussed later in the results section.

Moreover, since this is a structure-based study, I would expect the authors would discuss further the breakthroughs in structural studies of nAChRs that have paved the way for such studies.

There is a wealth of information presented and the choice to merge the "Results" and "Discussion" sections makes difficult to follow the significance of the study. In my opinion, a clear distinction of the two sections will greatly improve the readability of the paper and eventually the work will be more cited by the community.  

Their methodology sounds solid.

A question that will first come to the mind of every nAChR expert is why the authors used the humanized AChBP chimaera instead of the 1.5 year old cryoEM alpha7 structures? An explanation about that choice should be provided, especially since the resolution of the alpha7 cryo-EM structures is very well and significantly better that the alpha4beta2 structure (5kxi). 

Lines 244-246 should be included in the figure 3 legend. 

The second, large sentense (lines 413-417) of the "Conclusions" section should be written in better english and probably should be splitted. 

Reviewer 2 Report

The manuscript entitled “Pharmacophore mapping combined with dbCICA reveal new 2 structural features for the development of novel ligands target- 3 ing α4β2 and α7 nicotinic acetylcholine receptors” reported the insight of ligands target- 3 ing α4β2 and α7 nicotinic acetylcholine receptors through molecular docking and pharmacophore modelling. The manuscript is well explored and well written. Following a few things are required before further evaluation.

1.       The introduction is very brief. I would like to recommend to re-write the introduction explaining the clear objective of the study.

2.       The details of receptor selection are missing. The authors must explain details about the procedure of receptor selection.

3.       It is highly recommended to perform the MD simulation for at least 100ns of top 5 molecules bound with both receptors.  Insight of the molecular interactions through MD simulation must be discussed.

4.       The binding energy must be calculated using MM-GBSA/MM-PBSA approach.

5.       A clear future prospect of the work must be given.

Round 2

Reviewer 2 Report

The manuscript may be accepted for publication.